# The Psychological Distress and Quality of Life of Breast Cancer Survivors in Sydney, Australia

**DOI:** 10.3390/healthcare10102017

**Published:** 2022-10-12

**Authors:** Laura-Anne Aitken, Syeda Zakia Hossan

**Affiliations:** Faculty of Medicine and Health, The University of Sydney, Camperdown, NSW 2006, Australia

**Keywords:** breast cancer, psychological distress, quality of life, women’s health, qualitative study

## Abstract

In Australia, breast cancer is one of the most common cancers affecting women. Between 1987–1991 and 2012–2016, the five-year survival rate improved from 75% to 91%. The increased chance of survival due to early detection and treatment interventions has resulted in more women living with the diagnosis. This qualitative study was designed to analyse the journey of breast cancer survivors, their experience of psychological distress and changes in quality of life (QOL) due to the increased prevalence amongst Australian women. In-depth interviews were conducted; they lasted over 45 min and comprised 15 participants. The main topics discussed were knowledge of breast cancer prior to diagnosis, psychological distress, QOL and experience of use of healthcare services. The results showed that the process of diagnosis, undergoing treatment and isolation post-treatment resulted in high amounts of psychological distress. A reduction in QOL was also experienced due to treatment and medication side effects, fatigue, cognitive changes, and body-image perception. These findings can assist researchers in providing evidence-based frameworks for policy changes and for further investigation into effective healthcare interventions.

## 1. Introduction

Breast cancer is one of the most common causes of cancer-related deaths in women globally. It occurs when abnormal cells in the breast grow uncontrollably. It is more common among women; however, a small proportion of men also are affected by breast cancer [1]. After diagnosis, some common treatments that Breast Cancer Survivors (BCS) receive are a lumpectomy, mastectomy, chemotherapy or hormone therapy [2]. Due to the raised awareness and focus on early detection programs, survival is becoming increasingly likely for breast cancer patients [3]. Within Australia, breast cancer in 2016 was the most common cancer affecting women, with 330 new cases per 100,000 people [4]. Between 1987–1991 and 2012–2016, the five-year survival rate improved from 75% to 91%, increasing the number of survivors in the community [4]. This is due to increased awareness campaigns and breast screening availability provided by companies, including BreastScreen Australia [5]. BreastScreen Australia is the state and territory initiative of the Australian government to increase early detection and thus decrease illness and death. It allows women over 40 to have a free mammogram every two years [6].

For BCS, common issues range from treatment side effects [5], financial hardships [7] and comorbidities [8] to social isolation [9]. Many studies have assessed the health related QOL of BCS [5,10,11,12]; however, there is limited research that focuses on Australian women who have survived cancer and continue to experience its detrimental effects. QOL and psychological distress are essential patient-centred outcomes, assisting with assessing cancer care delivery. BCS is a medical and social label for all who live with cancer diagnosis until death, irrespective of the cause [13]. These BCS undergo considerable stress and trauma throughout diagnosis, treatment and post-treatment, which continues even after survival, subsequently impacting their QOL. The main factors identified in the literature as the significant causes of stress and trauma include psychological distress, financial stress and social isolation.

Studies have indicated that BCS experience mental health issues as soon as they are diagnosed with cancer, during treatment and survivorship. Two main concerns within psychological distress have been identified for BCS: mental health issues and distress surrounding cognitive function. One-fourth of breast cancer patients will develop anxiety and depression at some point in the breast cancer journey [14]. Those aged younger than 50 years are especially likely to report psychological distress [15,16] compared to the older cohort aged > 50 years. However, it is also said that 12 months post-diagnosis, most BCS have returned to pre-diagnosis levels of distress [17]. This discrepancy between studies highlights the importance of a study to outline mental health issues throughout the journey, rather than just certain checkpoints.

QOL is an individual’s perception of their position in life, in the context of the culture and value system they live within along with their goals, expectations, standards and concerns [18]. It is clear from the review of the literature that various stressors have resulted in a reduction in QOL for BCS. Previous research on QOL has focused on understanding the QOL in the early phases of breast cancer and groups of older BCS [10]. Most of these studies, however, focused on survivors 10 years post-diagnosis. The conclusions drawn were similar, with the participants reporting low QOL with additional sexuality, pain and psychological distress issues. This was illustrated in a quantitative study with BCS between the ages of 40 and 49, which found that the presence of breast-related symptoms at the time of survey completion had a profound impact on QOL [5].

Within the literature, there is insufficient focus on the qualitative experience of Australian BCS from diagnosis to survivorship. Most of the literature available is based on quantitative studies, with limited research being available on the experiences of BCS. This research is an attempt to fill this gap. There has also been a focus on mental health and cognitive disorders such as struggles with depression and increased anxiety; However, there is a need for a greater understanding of the progression of these issues in Australian BCS throughout their entire BC journey.

This research aims to understand the journey undergone by BCS diagnosed with breast cancer from diagnosis to survivorship. Furthermore, it explores the challenges experienced by the BCS due to physical changes, financial hardship, emotional distress and social isolation The Australian healthcare system is unique due to the country’s cultural diversity and different demographics. Understanding the experience of specifically Australian women with breast cancer is essential, as it will allow healthcare providers to understand the experiences of BCS in greater depth within the cultural context. This, in turn, may help facilitate greater sensitivity when treating their patients and assist in developing strategies and policies for BCS.

This cross-sectional exploratory research is based on a qualitative research method. It aims to understand the journey undergone by women, diagnosed with breast cancer, from diagnosis to survivorship. Further, it explores the challenges experienced by the BCS due to physical change, financial hardship, emotional distress and social isolation.

## 2. Materials and Methods

Eligible participants were BCS; they were aged 35 years and above, living in Sydney and had been diagnosed for at least a year. A total of 15 women with breast cancer were recruited for the study. Participants were excluded if they had been diagnosed less than one year ago to ensure all participants had completed their treatment. Participants that were diagnosed greater than 11 years prior were also excluded as they may not clearly recollect their early experience of diagnosis. Multiple strategies were used to recruit participants including snowball and convenience sampling, social media and community organisation advertising. For example, through the assistance of the organisation Pink Hope, a preventative health charity, we acquired participants using flyers. Advertising on social media gained the most traction, with participants sharing and passing on information to other survivors. Before recruitment commenced, ethics approval was obtained from the University of Sydney Human Research Ethics Committee (25 August 2021). The participants were also presented with the “Participant Information Statement” to ensure the study had informed consent and that participation was entirely voluntary and anonymous. The data were also saved in OneDrive and password-protected, and only researchers had access. These in-depth interviews were conducted over 45 min with 15 BCS. The first author participated in two days of training prior to the interviews commencing.

This specific qualitative method was chosen to ensure that the full experience of BCS was explored effectively. In particular, QOL was measured subjectively rather than using established measurement tools. Quantitative measurement of QOL qualifies experience’s on a scale rather than exploring participants’ experiences which was important for this study. In the in-depth interviews, the interviewer led the BCS through a series of open-ended questions to establish the experience of breast cancer diagnosis, treatment and living with the diagnosis (Appendix A). The interviews were conducted over 3 weeks in August and September 2021 and were recorded via Zoom. These in-depth interviews allow the participant to determine the direction of the interview and are, therefore, an effective method for an exploratory study [19].

A thematic analysis was conducted by researchers on the interviews [20]. Firstly, each interview was transcribed. An excel document was then prepared with each of the interview questions as titles. The responses from every participant relating to that question were then compiled for easy comparison. The first author then went through each response and highlighted common words and phrases used by the participants. These words and phrases were then grouped together to form the codes for each question. These codes were used to identify the common themes and to identify similarities between the BCS journeys.

## 3. Results

The main themes identified within this study were emotional and psychological distress throughout the journey of BCS, a reduction in QOL seen through social changes and physical symptoms and access to health services.

### 3.1. Background Information

#### 3.1.1. Sociodemographic Background

The sociodemographic background of the study participants is presented in Table 1. The average age of the study participants is 45 years, the youngest being aged 35 years. Participants had received their diagnosis between 1 and 11 years prior to the study. On average, 66% of participants were employed with education levels ranging from year 12 graduation to tertiary studies. The majority (80%) of participants were married and had between 1 and 4 children (See Table 1).

#### 3.1.2. Previous Knowledge of Breast Cancer

Participants were asked to recount the knowledge they had of symptoms, diagnosis and treatment of breast cancer prior to their diagnosis. Three participants had a family history of breast cancer and were more likely to be informed about breast cancer. Symptoms identified by the participants were lumps, swelling, sensitivity and pain in the breast. Around 66.7% of participants had heard of a mammogram before, and 40% knew of breast self-examination (BSE). The most common treatment identified was chemotherapy, with 73.3% reporting prior knowledge (see Table 2). Some of the comments by the study participant’s symptoms, diagnosis and treatment of breast cancer are presented in Table 2.

#### 3.1.3. Diagnosis of Breast Cancer

Participants were asked, “How did you find out you had breast cancer?” The three main methods of diagnosis were finding a breast lump through self-examination, noticing changes in their breasts and through ultrasounds and mammograms. Approximately 40% mentioned that, initially, they dismissed the symptoms as being hormonal or something less serious (See Table 3).

#### 3.1.4. Treatment of Breast Cancer

Of these 15 participants, the most common sources of treatment reported were chemotherapy, radiation and surgery. Out of the total participants, 86.6% had chemotherapy and 53% had radiation, and all the participants underwent surgery (the majority received a mastectomy 66.6%). Side effects from these treatments and the preventative medications were also highly reported to impact QOF (See Table 4).

### 3.2. Psychological and Emotional Distress

The study participants were asked to share any experience they had of psychological or emotional distress. Open-ended questions were used to assist participants in outlining their experience during diagnosis, treatment and post-treatment. Responses were categorised under the broad themes and presented below:

#### 3.2.1. Experience of Receiving Diagnosis and Treatment of Breast Cancer

Most participants (46%) went into a state of shock when they received their diagnosis, with three of the fifteen participants reporting a memory loss or blurred memories of the moment and four others reporting a trauma response (hot flushes, anxiety).

Participant 010 described the experience as:

“My whole world just collapsed. And I think I just sat in his room for like 20 min wailing, feeling like I was gonna be sick.”

Three of the fifteen participants reported crying, while four mentioned moving into “survival mode” to attack the problem pragmatically. Participant 008 stated that: 

“First question was, out of my mouth was what are we doing about it? Tell me what to do next…I was very practical, very pragmatic.”

#### 3.2.2. Treatment: Chemotherapy, Surgery and Radiation 

Of the 15 BCS interviewed, 13 went through at least one round of chemotherapy, either pre- or post-surgery. About 40% of participants reported a feeling of losing their identity, and 33% reported feeling helpless. For example, participant 015 stated that:

“Chemo is terrible ’cause you lose not just your hair, you lose your eyelashes and your eyebrows, you lose absolutely everything…you just go grey, and again, it is the treatment that makes you sick… I found it overwhelming at times, I was quite emotional.”

#### 3.2.3. Post-Treatment

After treatment had been completed, several BCS reported a lack of direction and support, which caused psychological distress. Due to the continuing side effects of medications, menopause and the abrupt nature of ending treatment, 40% of BCS report of feeling abandoned and isolated.

“I have had such a long time of treatment and now, I feel like I am going into this open water of not as much intensity.” (P010)

Participant 011 also described the inability to deal with long-term medication side effects that caused considerable distress in many BCS.

“There is always an end in sight…you have the surgery, and then you want clear margins, and you get your diagnosis…You know how many treatments of chemotherapy you are having over a period of time, You know how many treatments of radiation you are having over a period of time. So, once you get through all of that they put you on this medication, which the side effects don’t end…There is no end in sight.” (P011)

#### 3.2.4. Anxious for Recurrence

A common phrase repeated amongst participants was “scanxiety” (3 participants), which describes the feeling of panic and stress before a check-up and scan. BCS also report a hyper awareness of aches and pains and how they jump to the conclusion of cancer quickly (46%). Four BCS described this anxiety as always being in the back of their minds and as constant stress. Participant 008 especially struggled with this stress and described her experience:

“Biggest thing that I battle with all the time, it’s always there. It may not rear its ugly head every day, but it is always at the back of your mind, always.” (P008)

### 3.3. Quality of Life

Open-ended questions were asked about the impact of breast cancer on the QOL of the study participants. These questions focused on daily activities, sleep, fatigue, cognition, physical body image and sexuality. The three main themes that contributed to changes in QOL were physical symptoms, support network issues and finally accessibility to services.

#### 3.3.1. Physical Symptoms

Several common symptoms impacted the QOL of BCS. These included the inability to sleep, increased fatigue, chemo fog, vaginal dryness and changes in body image. Almost all BCS reported having issues with the perception of their bodies. The majority reported embarrassment of their bodies, particularly how they look in front of others, feeling unattractive to partners and a reduction in self-esteem. Five of the study participants highlighted concerns with the reconstructions, while three BCS outlined how not having nipples contributed to this issue. Participant 006 linked her issues with body image to her relationship with her husband and her sexuality.

“Now it is just very obvious I don’t have nipples, but they are just it looks like I’ve been massacred. And my husband is a bit sad about it all [chuckle]. Unfortunately, men are very…They’re quite visual creatures, unfortunately, in my experience, from what I’ve seen, but he’s very understanding, and I think it’s more the problem of the menopause, which is why sex life is dwindling.” (P006)

Some BCS did report that cutting off their own hair before it fell out was a way to regain some control.

“I did make a pre-emptive strike to cut off my hair, but not straight away, ’cause I used to have really long, curly hair, and I thought, Well, that’s something…Like I think a lot of women, something you can control.” (P007)

Difficulties sleeping were reported by 66% of the BCS, while the other four participants had no issues with sleeping. Four BCS mentioned that hot flushes and side effects from menopause are the reason for this poor sleep, whereas two others associated poor sleep with an overactive mind. Three BCS use sleeping tablets and melatonin to get to sleep.

“So not very good sleep to begin with and the chemo, I couldn’t sleep, I couldn’t…I just felt sick the whole time. Trying to sleep when you’re feeling nausea, wasn’t fun, so it was very difficult trying to sleep and then getting up and smells really affected me.” (P014)

Fatigue not only impacts BCS during chemotherapy but lasts for years post-treatment. Three women reported that they feel like they have less energy, and two others say they have issues with maintaining work-hour requirements. On average, fatigue was worse during treatment, and now most BCS have reported improvements. Participant 009 was one of many BCS who experienced issues returning to work due to fatigue. 

“I would say for the first three years the fatigue is really, really tangible, very hard to, I guess, to sustain things for a long period of time…I found working, going back to work, even though it was a couple of days a week at first, were exhausting. Now, I still, even now, I still don’t have the physical, I guess, stamina that I used to have.” (P009)

Cognitive changes impact self-confidence and workability. BCS have reported feeling like they can’t remember words and suffer from chemo fog and short-term memory loss. Four of these participants are worried about these cognitive changes impacting their working life. This issue was highlighted by participant 002, who stated:

“As well, so I think once I got back into that, I felt like my brain came back online, but before that, It’s like I could. Sometimes I couldn’t even find the words to say, and I was worried. I’m like how am I gonna go back to work and have these conversations with families? And you know, I sound dumb pretty much.” (P002)

Bodily changes due to menopause, medication side effects and changes in self-esteem have been seen to reduce BCS ability to be intimate. For varying reasons, from the experience of dryness and pain to a lower libido, 60% of the participants reported some change in their sex lives. Participant 009 describes her struggle with getting hormone replacement therapy to reduce symptoms and improve her ability to be intimate.

“I had a very healthy sex life with my husband before any of this happened. Now it is non-existent. Not fun, not wanting to. But it is extremely painful. It doesn’t…I can’t obviously, can’t take HRT, I have tried. I have begged, I have screened, I have fought with doctors, I’ve been shut down at every turn.” (P009)

Another participant outlined the steps she has taken to improve her sexual functioning.

“Firstly, you don’t really feel like it, that’s for sure. But then afterwards, I think because of menopause, you’ve dryness and discomfort, definitely suffering from both of those. But because of my involvement volunteering with The Mater, I was lucky enough in those groups…So, she talked us through the option of laser in your cervix to get that sort of moisture back. But I…So we need to use lubricant now because it’s never really got back to 100% normal.” (P008)

#### 3.3.2. Support Systems

Support system changes were identified in the breast cancer survivors’ relationships with their children, husbands, friends and family members. These changes not only impacted them emotionally during the time of treatment but have also had lasting effects post-treatment. Most of these BCS (80%) were married and had children, and a majority of them reported that their husbands were extremely supportive.

“He was very supportive. I was very lucky. I you know on being on breast cancer support group pages that I so many stories of relationships breaking down and you know divorce and all the rest of it. But yeah, no I was very lucky he was 100% in it with me.” (P002)

A few BCS did outline issues with children’s behaviour or difficulties amongst the extended family unit. Participant 014 in particular noted the difficulties within a family unit. 

“My little girl was quite young at the time, so she didn’t really know what was going on, but her behaviour started becoming quite bad during it all…So it was difficult for the older kinds, but they all worked so they threw themselves into work and study and friends and kind of like, I felt like they ignored me a lot.” (P014)

Seven out of the fifteen participants’ social networks shifted during their journey. Three BCS highlighted their desire for privacy throughout this time, while three others mentioned that they felt forgotten. Six BCS specifically mentioned that they felt disconnected and distant. Alternatively, 33% of BCS reported not feeling isolated and completely supported by their community, family and friends

For participant 008, it came as a surprise as to who stepped up and who stepped back.

“I felt like my network went into the washing machine and some people went down the drain, and others who came out lovely and fluffy; they are my posse. I’d trust my life with them. It’s quite amazing the people who step away…But I had a couple of very, very close friends who are not in my inner circle anymore, and just stepped away.” (P008)

### 3.4. Health Services Experience

BCS in both the public and private sectors reported financial distress throughout their journey (see Table 5). Out of the 15 participants, only 5 reported not having financial stress during this time. Of these BCS, all five were part of the private health system. Many did, however, acknowledge the steep prices and shock felt over the cost of their treatment, commenting that they were “lucky” to be able to afford treatment. Of the BCS experiencing financial stress, three were in the public system, and five were private. One woman highlighted the stress of having a lack of income on her family:

“My husband had to take time off because he had to return to his work. The cancer therapy helped us pay half the bill, which they do because of the stress that comes on between a family and then the kids trying to go to school, go to work, and then your husband’s gotta look after the little one, cook, clean and wash.” (P012)

There were also reports that post-treatments, due to financial constraints, BCS were not able to access services to improve their QOL.

“Everything that’s come after it is virtually out of pocket so that I can’t afford to seek the therapy that I actually really need on a regular basis.” (P007)

During treatment, interactions with the medical staff and ease of using services contributed to the overall QOL of BCS. One woman mentioned that she felt “like everyone was treating me like a queen. It was great. I’m like everyone so nice to me because they know I’ve got cancer” (P002), whereas another mentioned that during treatment she had a few issues with communication that led to further anxiety.

“Oh, you’ve got more of a chance of it coming back somewhere else than in the other breast”, and I remember thinking, “Did you just hear what you just said?” They just…I know it’s hard for them and they need to be separate emotionally, but they also need to remember that they’re dealing with people and they’re dealing with people’s lives”

As mentioned previously, there was a drop-off in support after active treatment. Here, BCS reported that they had difficulty getting help for the side effects of medications.

“They don’t give you any way to manage those symptoms of what you’re gonna go through on the medication. They don’t identify things that you can go, “Oh, that’s what that is, and this is how I can treat it, how to avoid it.” They just kind of ignore the fact that you have side effects from it, or they say, “You know, I’ve seen people worse off, so you’ve got it okay.” (P011)

## 4. Discussion

This study revealed that participants frequently reported psychological distress and a reduction in QOL. This aligns with the previous literature, where issues of financial hardships [7], comorbidities [8] and social isolation [9] were reported. The most prevalent issues highlighted by these studies BCS were the harm of the side effects from post-treatment medication, anxiety around cancer recurrence and issues with health service utilisation.

Psychological distress is the emotional suffering associated with stressors and demands that are difficult to cope with daily. The results from this study illustrate that the significant points of psychological distress along the journey of BCS are receiving the diagnosis during treatment and, finally, the process of survivorship. The most commonly endorsed stressors include chemotherapy trauma (46%) and loss of identity (40%). The anxiety of reoccurrence was noted by all survivors, with three of the fifteen participants reporting that it no longer concerns them. The literature has observed that BCS levels of anxiety and depression decrease over time, which contradicts the findings of our study where it has increased [21]. The increase in distress during the transition from active treatment aligns with qualitative and anecdotal studies [22], whereas contradicting quantitative studies report a decrease in distress at this time [23,24]. There were three main events where psychological distress occurred for BCS. These included receiving the diagnosis, completing treatment and post treatment abandonment which have also been observed in other studies [25,26]. When diagnosed, participants went into a state of shock and experienced trauma responses. Four of the fifteen participants did, however, mention that their first instinct was to attack the problem pragmatically. The next source of distress was going through treatment, especially chemotherapy. Side effects such as losing hair and the feeling of helplessness were significant contributors to psychological distress. Finally, post-treatment feelings of abandonment by the healthcare system and the lack of direction also contributed to distress, which is supported by a growing body of evidence [27,28]. Here, it was evident that participants felt abandoned and struggled with not having any support networks to assist with the transition.

Participants’ QOL was explored by discussing how certain indicators, identified from the literature review, have impacted participants’ QOL. The significant concerns derived from these interviews were physical symptoms of BC, changes in support systems for BCS and poor body-image perception.

Poor body image is associated with mastectomies and breast reconstruction surgery [11]. Although a smaller sample was used, this study supported those findings. Over half of the BCS reported a drop in self-esteem, feeling unattractive to partners and embarrassed by their breasts. There was a consensus amongst two of the BCS claiming to be unsatisfied with their reconstruction, often suggesting that they struggle with having or not having tattooed nipples. Survivors have reported sexual health concerns for up to 10 years post-treatment [29]. In Pumo et al.’s (2012) study, sexual conditions that directly correlated to psychological disorders were prevalent in 36% of their participants [30]. Within this study, sexual issues and complications such as dryness, pain and low libido to side effects of treatments were linked with medications rather than psychological disorders. There were also reports of participants’ embarrassment of how they look affecting their desire to be intimate with their partner.

Sleep and fatigue are both factors contributing to the reduction in QOL. Several BCS reported insomnia, trouble staying asleep and extreme fatigue throughout the day. There was a common trend of fatigue and sleep issues being more prevalent during treatment and then improving as time went on, supported in the literature [31]. There was, however, a few BCS that reported persistent trouble sleeping and fatigue even years post-treatment. In Beck et al.’s 2010 study, this trend was also seen where some survivors continued to experience problems with sleep many years later [32]. This constant fatigue has resulted in reducing the workability of survivors and the ability to spend time with families, which has led to a reduction in QOL.

Cognitive changes were linked to memory loss and reduced capacity for work for the majority of the study participants. Similar findings were reported in previous studies, where participants noticed language, memory, spatial ability and motor function changes [33]. However, there was some discrepancy among participants regarding whether the medication, treatment or effects of ageing were causing the changes in cognition. Collins et al. (2009) discussed this phenomenon concerning hormonal therapies such as tamoxifen and anastrozole [34]. It was concluded that hormonal therapies subtly have a negative influence on cognition. This aligned with our findings as the negative side effects of tamoxifen were mentioned by four of the fifteen participants and linked to their cognitive changes and other physical symptoms. There were also conversations linking these cognitive changes to stressors in the workplace, with BCS worrying that it would affect their performance. Similar findings were cited in Nelson and Sul’s (2013) study, which noted that the detrimental effects of reduced memory and learning impacted the workplace and function roles [35].

The majority of participants were married and had children. This resulted in extensive conversations about relationship changes and social isolation. Most BCS reported that their husbands were highly supportive and a great help during this time. Only a minority mentioned relationship breakdowns and increased stress due to their partners. Multiple studies have illustrated the positive effect of supportive spousal relationships [36,37], which our study supports. Even with this support, many BCS still feel isolated post-treatment due to a lack of continued understanding of their experience from family and friends [38] and continued side effects of medications. Perz et al. (2013) explored that the heightened fear of rejection experienced by BCS with potential partners causes survivors to isolate themselves even further [39]. The single woman who participated in this study reported that they felt embarrassed by their breast appearance; however, this fear of rejection was not explicitly mentioned. There was also no mention of participants further isolating themselves as they sought out more connections via social networks, support groups and their own families. Those who had better support networks of friends, family and communities mentioned that going through treatment was easier with all the extra help, and thus they felt less isolated. When this community was not present, there were often reports of feeling helpless and overwhelmed by tasks. This is supported in the literature, where those who have strong relationships and communities to assist them felt protected from negative feelings such as depression, anxiety and the stigma of BC [9,36,38].

The experience of using and accessing healthcare services was identified as a significant factor contributing to QOL. Within Australia, Medicare is the publicly funded universal health care insurance scheme that all Australians can access. In addition to this, private health coverage is also available to paying customers to cover the gaps in Medicare. These two services however still do not alleviate the entire financial burden of Breast Cancer treatments and services. Financial distress acted as a significant barrier to healthcare services for BCS [40]. This result was expected because various studies outlined expensive costs and lack of coverage via Medicare and private health [40]. The previous literature also outlined that breast cancer comorbidities would also be a barrier to returning to work [41], which was also seen in this study. One central area discussed was the difficulties affording services post-treatment. Through the public and private health sectors, most women received some financial support.

In some cases, participants had to pick and choose the most needed services at a specific time rather than having access to all services that may improve their QOL. Although the majority did report that financially they were stable, all acknowledge the exorbitant costs of life-saving and life-enhancing treatments during survivorship. Breast reconstruction was a significant source of financial strain. It is not covered under Medicare, and as it is not considered a life-saving surgery, private health services often do not cover it.

Treatment and communication by staff within the healthcare system had a significant impact on survivors’ experience and distress. Four of the fifteen BCS reported at least one issue of switching practitioners or feeling like they were not being heard. One of the most considerable contributions to distress during diagnosis was the communication of medical staff about the severity and course of action for treatments. Participants also highlighted how their health was now in their own hands after treatment, and there was no transition phase. Many felt abandoned and isolated by the shift in control as well as being at a loss as to how to manage their symptoms into the future. Most participants continued to take medication after their treatments to prevent cancer recurrence. These medications, however, had harmful side effects that BCS have had to learn how to manage by themselves. This, paired with financial restrictions, has resulted in a reduction in QOL.

However, there are a few limitations of the study. The limitations of this study include social media and readiness bias. The effectiveness of social media recruitment resulted in a biased sample toward BCS who do not use or do not have access to social media. In addition, it also eliminates participants who are not yet ready to share their experiences as they are still dealing with the consequences of their disease.

## 5. Summary and Conclusions

Fifteen BCS were interviewed to understand their journey from pre-diagnosis to survivorship. The average participants were aged 35–45, had at least one child, were married and were Australian. The most common issues mentioned throughout this study were the lack of interventions for medication side effects, changes in their body image and anxiety around cancer reoccurrence. There were also many accounts of how physical side effects from treatments have impacted their work and social and sexual lives.

These findings are based on a qualitative study, with a small sample of 15 in-depth interviews; therefore, a generalisation of the study cannot be made. Due to the small number of participants and sampling error, there was also a lack of older participants from diverse backgrounds.

This study has provided a preliminary basis for the experience of BCS and can expanded on in a larger study using quantitative methods to assess QOL and psychological distress. These findings have implications for policymakers and for the healthcare industry along with the importance of a survivor’s relationship with their healthcare team and their ability to access services, which was especially highlighted. It would be our recommendation to focus future studies on the experience of BCS from ages 35 to 45 including their diagnosis. Future studies should also include BCS of culturally diverse groups and older age cohorts.

From this research, it is also clear that younger women should have access to free and convenient mammograms. Additionally, the experience of sexuality and early menopause is an area that needs to be examined to provide better services and support for survivors. Access to post-care services, links to support groups and information about medication side effects should be prioritised in order to increase survivors’ QOL.

## Figures and Tables

**Table 1 healthcare-10-02017-t001:** Sociodemographic Background.

Variables	Subgroups	% (n)
Age (average = 45)	35–39	26.6 (4)
	40–45	20(3)
	46–49	6.6 (1)
	50–55	13.3 (2)
	56–60	13.3 (2)
Employment Status	Fulltime	20 (3)
	Part-time/casual	46 (7)
	Unemployed	33.3 (5)
Level of Education	High School	26.6 (4)
	Tertiary studies	
	- Diploma	26.6 (4)
	- Graduate Certificate	6.67 (1)
	- Undergraduate	13.3 (2)
	- Postgraduate	26.6 (4)
Religion	Christian	66.67 (10)
	No Religion	26.6 (4)
	Muslim	6.67 (1)
Marital Status	Married	80 (12)
	Divorced	6.66 (1)
	Single	6.67 (1)
	De-facto	6.67 (1)
Family Income	20,000–50,000	13.3 (2)s
	50,000–100,000	26.6 (4)
	100,000–150,000	6.67 (1)
	150,000–200,000	26.6 (4)
	200,000+	20 (3)
Children	0	6.67 (1)
	1–2	53.3 (8)
	3–4	33.3 (5)
Country of Birth	Australian	80 (12)
	Argentina	6.67 (1)
	India	6.67 (1)
	Bangladesh	6.67 (1)

**Table 2 healthcare-10-02017-t002:** Breast cancer previous knowledge.

Q: What did You Know about the Symptoms of Breast Cancer?		Q: What did You Know about How to Diagnose Breast Cancer?		Q: Did You Know How Breast Cancer was Treated?	
Example Responses		Example Responses		Example Responses	
“If you feel a lump or anything abnormal sort of on your chest or under your armpit, sort of self-examining that way” (P 005)		“My mother had had breast cancer…GP was very conservative…I would walk in and have a mammogram and an ultrasound every year since I was 40” (P015)		“Well, I knew obviously, chemotherapy was gonna be usually…Other than surgery, chemotherapy is usually the first medical treatment after surgery, and then I knew, obviously, radiation as well. I didn’t know all the in-depth parts of having to get set up for all those treatments, but I knew about those treatments” (P011)	
As I always had thought, it would be like a lump. I was always had the breast cancer described as being a lump, like a pea or like a stone” (P007)		“I hadn’t done any mammograms or ultrasounds before. And to be honest, I’d never checked myself really, ever. I think I maybe did a couple of times, but it wasn’t something every month like you’re supposed to. So yeah, I didn’t do that” (P010)		“Well, I thought it was literally just either surgery or chemo like that two options, that’s really it. But didn’t realise how much options within each of those there were as well, yeah” (P003)	
		**Common Phrases % (n)**			
Lump	66.67 (10)	Self-Check	40(6)	Surgery	33.3 (5)
Changes	13.3 (2)	Dr. Check	20 (3)	Mastectomy	20 (2)
None	20 (3)	Mammogram	66.67 (10)	Chemotherapy	73.3 (11)
		Ultrasound	20 (3)	Radiation	46.67 (7)
		Biopsy	6.67 (1)	None	20 (3)
		None	13.3 (2)		

**Table 3 healthcare-10-02017-t003:** Experience of Breast Cancer. How did you find out you had breast cancer?

Keywords	% (n)	Responses
Self-check to find lump	46.67 (7)	“I actually felt and saw the lump. I was just doing a little, not spring clean, but just moving little things around and yes, found a lump form, tried to massage it, didn’t go away and sort of went to my GP to check it out” (P005)
Changes in breast	33.3 (5)	“I just woke up one morning and had a swollen breast, one swollen breast that was a little bit tender. [chuckle] And it was a little strange, it was different for me, but it was also something that I was going to quickly dismiss and pretty much did really, ’cause it went away within two days” (P003)
Pain	20 (3)	“I was in the shower one morning and I kind of leant over to wash myself and I felt a pain kinda of in the side of my breast on the left, then I felt there and there was a distinct lump” (P009)
Dismissed symptoms (hormonal)	40 (6)	“Anyway, I had a…Early in the year, I did go and see my GP to say, ‘Something really doesn’t sort of feel quite right’, but I was completely just putting it down to hormonal, to relative to hormonal changes” (P007)
Fast diagnosis and movement to treatment	33.3 (5)	“Went to the doctor straight away. So I think within two days, I had an appointment with the doctor. From there, I went and had a mammogram and an ultrasound. He got me into the specialist very, very quickly, I had a biopsy at the specialist, I saw the specialist, I had breast cancer. So within…I think within probably two weeks” (P009)
Mammogram + ultrasound	60 (9)	
Misdiagnosed/long diagnosis process	20 (3)	“Doctor on board Queen Victoria said, ‘I think you’re crazy, you don’t need to go back. I thought you’ve already been cleared, you don’t need a biopsy’…took them two years for them to diagnose me on board Queen Victoria, and by then I’d seen five of their doctors and I had reported with each of the doctors saying, ‘Oh, this couldn’t possibly be any problem, and it couldn’t possibly need a biopsy. She’s just too young.’” (P006)

**Table 4 healthcare-10-02017-t004:** Treatments % (n).

Mastectomy	80 (12)
Chemotherapy	93.3 (14)
Radiation	53.3 (8)
Reconstruction	26.67 (4)
Lumpectomy	13.3 (2)
Oncoplastic Resection	6.67 (1)
Hysterectomy	20 (3)

**Table 5 healthcare-10-02017-t005:** Financial Stress and Healthcare System.

	Financial Distress n (%)	No Financial Distress (n)
**Public System**	3 (20%)	0
**Private System**	5 (33%)	5 (33%)
**No comment**	2 (13%)	

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
