# Peer review of "The Psychological Distress and Quality of Life of Breast Cancer Survivors in Sydney, Australia"

_healthcare, 2022, doi:10.3390/healthcare10102017_

Round 1
Reviewer 1 Report
Please see review attached.

Author Response
- “It is more common among women; however, a small proportion of men also get breast cancer”> Since you add this section, it would be more inclusive to then use gender neutral terms going forward with the rest of the paper when addressing BCS aside from instances that reference specific studies that focus on certain sociodemographic(s). - “For BCS, common issues range…”> Provide clarification that this is shorthand for breast cancer survivors at the first instance of using this acronym. This is also something that should be done with the quality of life acronym (QOL).
Response: Fixed throughout document (note track changes)
- When stating “cognitive disorders” when referring to the cognitive problems that BCS may experience, it would be a good idea to provide some examples of these cognitive complaints or disorders (cognition is a really broad topic). Otherwise, the only other time it is expanded upon is briefly in the results section.
Page 2 Paragraph 6 added: such as struggles with depression and increased anxiety
- “Within the literature, there is insufficient focus on the experience of Australian women from diagnosis to survivorship.” > Given the subsequent sentence, do the authors mean to state that there is insufficient focus in terms of *qualitative* research? Otherwise, the argument for the literature to focus more on *Australian women” is unclear.
Page 2 Paragraph 5: Within the literature, there is insufficient focus on the qualitative experience of Australian BCS women from diagnosis to survivorship.
- “The Australian health care system is unique due to the country’s demographic characteristics in terms of multiculturalism.” > Please clarify what “multiculturalism” means in this context. Please ensure that this follows for other instances in which “cultural context” is named.
2 Paragraph 6: The Australian health care system is unique due to the country’s cultural diversity and different demographics.
- Study aims (first paragraph) better fit at the end of the introduction. Similar, the last sentence of Materials and Methods seems more fit for the introduction or discussion, as it is not an objective description of the methodology.
Response: Moved paragraph to end of introduction, removed last sentence
- “Participants were excluded if they were less than 35 years old, not living in Sydney, and had been diagnosed with breast cancer for less than one year”> This probably doesn’t need to be reiterated again. Unless there is something inherently different about the exclusion criteria from the inclusion criteria, the exclusion criteria are assumed (given the inclusion criteria) and can be taken out for reader clarity.
Response: Removed from method
- “... snowball and convenience sampling”> provide slightly more explanation (e.g., the authors note the social media had the most traction, but that leads the reader to wonder what other approaches were used). This only needs to be a brief sentence or explanation in parenthesis. Relatedly, a very brief description of Pink Hope is needed for readers unfamiliar with the organization.
Page 3 Paragraph 1: Multiple strategies were used to recruit study participants including snowball and convenience sampling, social media and community organisations advertising. For example, through the assistance of the organisation Pink Hope, a preventative health charity, we acquired participants through flyers.
- “Firstly, the data was coded by the researchers using colour coordination…”> What exactly was being colour coordinated? Please provide further description of the qualitative data processing and analysis approach.
Page 3 Paragraph 4: A thematic analysis was conducted by researchers on the interviews [41]. Firstly, each interview was transcribed. An excel document was then prepared with each of the interview questions as titles. The responses from every participant relating to that question were then compiled for easy comparison. The researchers then went through each response and highlighted common words and phrases used by the participants. These words and phrases where then grouped together to form the codes for each question. These codes were used to identify the common themes and to identify similarities between the BCS journeys.
- “Participants were diagnosed with breast cancer between 1-11 years.” > Could this be clarified? The
- diagnosis year was briefly stated in the inclusion/exclusion criterion section in the methods but in this statement, it is unclear what is meant by diagnosed between 1-11 years. Does this mean onset of symptoms? Additionally, the data are not detailed in Table 1 and may be easier to understand in table format.
Page 3 Sociodemographic background: Participants within this study had received their initial diagnosis between 1–11 years ago.
Page 3 paragraph 1:Participants were excluded if they had been diagnosed less than a year ago as they will may be partaking in treatments. Participants that were greater than 11 years post diagnosis may be able to recollect their early experience of diagnosis.
- “Common symptoms identified by the participants were lumps and changes in the breast” > Provide examples of “changes in the breast” that the participants verbalized. Additionally, provide the statistics for these symptoms identified or state that they can be found in Table 2. Using the word “common” to qualify the symptoms isn’t as informative as an exact statistic.
Page 4 Previous Knowledge of Breast cancer: Symptoms identified by the participants were lumps, swelling, sensitivity and even pain in the breast
- For the headers of Table 3, make sure to list the responses as ‘Example Responses’. At first read, it is confusing as there are only 1 response shown for each category of key words from only one participant.
Changes on page 5 table 3
- “Due to the continuing side effects of medications, menopause, and the abrupt nature of ending treatment, there are multiple reports of feeling abandoned and isolated.” > Provide the exact statistic (i.e., average number of reports that listed feeling abandoned).
Page 7 paragraph 6: Due to the continuing side effects of medications, menopause and the abrupt nature of ending treatment, 40% of BCS report feeling abandoned and isolated.
- Please include sample size along with percentages and vice versa; this follows APA style and is particularly important given the small sample size. “Only three participants reported crying” > at risk of minimizing participants’ experiences, I suggest rephrasing (e.g., removing the word “only”)
Page 7 Paragraph 2: Three of the fifteen participants reported crying, while four mentioned moving into “survival mode” to attack the problem pragmatically.
Page 10 paragraph 2: The anxiety of reoccurrence was noted by all survivors, with three of the fifteen participants reporting that it no longer concerns them
- "However, within this study, there was a focus on younger women from ages 35–45 because of our social media recruitment strategy” > how does this compare with average age in the literature?
Response: The median age for women in Australia to be diagnosed with cancer is 62 according to the Breast Cancer Network Australia. However there have been some studies outlining how migrant populations in Australia are at risk from a younger age. To ensure inclusion of all cultures within Sydney we created the age range of 35 and above.
- “The results from this study illustrate that the significant points of psychological distress along the journey of BCS are receiving the diagnosis during treatment and, finally, the process of survivorship.”> It would be helpful if the authors briefly described the most commonly endorsed stressors in this study.
Page 10 paragraph 2: The most commonly endorsed stressors including chemotherapy trauma (46%) and loss of identity (40%).
- “There were three main events where psychological distress occurred for BCS: receiving diagnosis and completing treatment, which has also been observed in other studies” > Listing the third main event here would increase clarity, instead of listing it several sentences later.
Page 10 paragraph 2: There were three main events where psychological distress occurred for BCS. These included receiving the diagnosis, completing treatment and post treatment abandonment which have also been observed in other studies [24,25].
- “Nipple tattooing specifically was mentioned to be a source of distress by participants.” > Is this also seen in past literature? In general, it is helpful to connect portions of the discussion to the current literature. Can this be expounded upon to understand why specifically this is an area of distress (e.g., because of cost, look, and/or pain)? Also, please ensure that the statistical figures around this are listed in the Results, as there should not be new data presented in the Discussion.
Response: Discussed later in paragraph (4). Removed form here to increase clarity
- There are several instances of using “most” and “many” to quantify data from the study. Please remember to include supporting statistics in parentheses as appropriate and to provide context for the reader
Response: Gone through document (see track changes) and added in clarifying values
Reviewer 2 Report
The authors have done an excellent job addressing with a qualitative method the important point of view of patients.
I have only few suggestions.
- A more detailed description of the process used to conduct thematic analysis (with citation of similar studies) can be usefull for readers
- Please check the percentages of each variable in Table 1 (eg number of children)
Overall, I believe the paper is acceptable for publication
Author Response
- A more detailed description of the process used to conduct thematic analysis (with citation of similar studies) can be useful for readers
Page 2 paragraph 5: Updated method section
A thematic analysis was conducted by researchers on the interviews [41]. Firstly, each interview was transcribed. An excel document was then prepared with each of the interview questions as titles. The responses from every participant relating to that question were then compiled for easy comparison. The researchers then went through each question and highlighted common words and phrases used by the participants. These words and phrases where then grouped together to form the codes for each question. These codes were used to identify the common themes and to identify similarities between the BCS journeys.
- Please check the percentages of each variable in Table 1 (eg number of children)
Page 3 table 1: Edited numbers
Reviewer 3 Report
This is a very nicely presented, qualitative study of 15 women breast cancer survivors in Australia who were interviewed with a structured outline that was recorded, transcribed and analyzed for specific themes. Much of what the authors report has been reported elsewhere, but the comprehensiveness of the material from the same group of subjects makes the findings more interesting and compelling. The authors mention several findings that other studies of this kind have not reported. They may wish to highlight and reinforce those findings because I think they are likely to have been correct.
Overall, a nice paper that I very much enjoyed reading.
I believe that the instructions to the reviewers could be made clearer by distinctly identifying each step in bold followed by specific instructions.
Author Response
We would like to thank you for reviewing our article submission.
Reviewer 4 Report
This is a qualitative study on the elements affecting the quality of life in breast caner survivors in Australia.
Breast cancer is a disease know to have important consequences on the lives of affected women especially since the progress in early diagnosis and effective treatment have substantially prolonged survivorship.
Regarding the methodology of the present study, 15 participants were recruited using also social media, hence the younger age of the sample. This is a limitation that should specifically be mentioned as it partly excludes women who do not use or do not have access to these media. Moreover, the sampling method results in a biased sample towards women who feel "ready" to share their experiences and who have probably has the opportunity of dealing with the consequences of their disease. In other words, the sampling method excludes women who are still coming to terms with the effects of breast cancer. I believe this should me commented in the text.
The time period of the interviews is not mentioned, neither is the number of interviewers used or whether they had previous training or not. These elements are important as they strongly related to the quality of the collected data.
There are numerous validated scales on measuring quality of life, yet the authors choose to interpret QoL subjectively. A comment on this choice should be made in the text.
Regarding the sample, do the authors have information on how many of the interviewed women were actually having screening mammograms prior their diagnosis? A brief description of how the breast cancer screening programmes are run in Australia would be useful to help the readers understand the context.
Moreover, do the authors have any information on time since breast cancer diagnosis when the interviews were taken? This could be an important piece of information to categorize the participants in relation to quality of life after diagnosis.
The insurance coverage policies is rather unclear. There are some mentions in the text on private versus state insurance, the term "being financially lucky" is used, reconstruction costs are not covered. It would be useful to shortly describe the coverage system in these options.
Finally, how is post treatment care organised in the country? Patient groups have been very effective in providing support to cancer sufferers. Is there a mechanism to streamline BCS towards these groups?
Author Response
- Regarding the methodology of the present study, 15 participants were recruited using also social media, hence the younger age of the sample. This is a limitation that should specifically be mentioned as it partly excludes women who do not use or do not have access to these media. Moreover, the sampling method results in a biased sample towards women who feel "ready" to share their experiences and who have probably has the opportunity of dealing with the consequences of their disease. In other words, the sampling method excludes women who are still coming to terms with the effects of breast cancer. I believe this should me commented in the text.
Page 3 paragraph 5 addition:
The limitations of this study include social media recruitment and readiness bias. The effectiveness of social media recruitment resulted in a biased sample toward BCS who do not use or do not have access to social media. In addition, it also eliminates participants who are not yet ready to share their experiences as they are still dealing with the consequences of their disease.
- The time period of the interviews is not mentioned, neither is the number of interviewers used or whether they had previous training or not. These elements are important as they strongly related to the quality of the collected data.
Page 3 Paragraph 3: The first author participated in two days of training prior to the interviews commencing
Page 3 Paragraph 4: The interviews were conducted over 3 weeks and were recorded via Zoom.
- There are numerous validated scales on measuring quality of life, yet the authors choose to interpret QoL subjectively. A comment on this choice should be made in the text.
Response: The quantitative measurement of QOL quantifies experiences on a scale rather than exploring participants experiences. This study aims to better understand the QOL from individual’s own perspective therefore we interpreted QOL qualitatively through participants responses.
- Regarding the sample, do the authors have information on how many of the interviewed women were actually having screening mammograms prior their diagnosis? A brief description of how the breast cancer screening programmes are run in Australia would be useful to help the readers understand the context.
Page 1 paragraph 1: BreastScreen Australia is state and territory initiative of the Australian government to increase early detection and thus decrease illness and death. It allows for women over 40 to have a free mammogram every two years [6].
Response: Yes, 6 of the participants had a mammogram prior to their diagnosis.
- Moreover, do the authors have any information on time since breast cancer diagnosis when the interviews were taken? This could be an important piece of information to categorize the participants in relation to quality of life after diagnosis.
Response: Yes, all participants are between 1-11 years since diagnosis. We have a record of the year in which each participant was diagnosed.
- The insurance coverage policies is rather unclear. There are some mentions in the text on private versus state insurance, the term "being financially lucky" is used, and reconstruction costs are not covered. It would be useful to shortly describe the coverage system in these options.
Page 11 Paragraph 8: Within Australia, Medicare is the publicly-funded universal health care insurance scheme that all Australians have access to. In addition to this, private health coverage is also available to paying customers to cover the gaps of Medicare. These two services however still do not alleviate the entire financial burden of Breast Cancer treatments and services
- Finally, how is post treatment care organised in the country? Patient groups have been very effective in providing support to cancer sufferers. Is there a mechanism to streamline BCS towards these groups?
Response: In some hospitals they provide the details of support groups and other services. This is a big disparity between public and private hospitals in the space of post treatment care. The more expensive hospitals such as Life House are reported to have better post patient care and access to such groups, other less funded hospitals have limited options for patients.